# Maternal Insanity in the Family: Memories, Family Secrets, and the Mental Health Archive

Alison Watts

Faculty of Health, Southern Cross University, Lismore, NSW 2480, Australia; alison.watts@scu.edu.au

**Abstract:** This work investigates my family's long-held secrets that concealed the whereabouts of my grandmother. After years of estrangement, my father discovered Ada living in a mental hospital. Memories are rarely straightforward and could only take us so far in understanding why Ada remained missing from our family for so long. My search for answers involved genealogical research and led me to access Ada's mental patient files. This rich data source provided some troubling glimpses into Ada's auditory hallucinations and grandiose delusions and her encounters with several mental institutions in Victoria, Australia, during the twentieth century. Critical family history approaches allow me to gain insights into the gendered power relations within her marriage and the power imbalance within families. The theme of migration is addressed through the lens of mobility when Ada relocated following her marriage and her movement between home on trial leave and several sites of care after her committal. Scholars have shown that the themes of migration and mobility are important and hold personal significance in exploring the connection between mental health and institutionalisation for our family. Here, I demonstrate how mental illness in families is stigmatised and concealed through institutionalisation and its legacy for younger generations.

**Keywords:** family secrets; family history; memories; maternal insanity; Australian psychiatric practices; institutions; mobility; twentieth century

## 1. Introduction

As a child, I found a small brown suitcase in my wardrobe with two silver latches and a squeaky wooden handle. 'Ada' is inscribed on the bottom in pencil, a name I did not recognise. I asked my mother: 'whose suitcase is this?' Mum replied, 'your grandmother's.' I was confused about who my grandmother was. I was born in the early sixties and grew up with my parents and two older brothers in Melbourne's eastern suburbs. Around the time of my birth, my father, having grown up without his mother, Ada, had discovered her living as a mental patient in Mont Park psychiatric hospital, Bundoora, Melbourne.

Ashley Barnwell (2019) found that emotional distance from deceased relatives involved in family's secrets can free up descendants from shame and allow the disclosure of their pasts. As Ada's granddaughter, it is likely that my place within the family as the youngest of Dad's offspring is distant enough from my parents' painful experiences so as to examine the secrets surrounding Ada's committal into a mental hospital. In this way, from my family position as an 'insider', self-reflexivity, and presence, as described by Hertz (1997) and Primeau (2003), become a critical feature in this research.

Mutual trust, cooperation, and rapport with family members can stimulate good anecdotes and family stories (Miller 1997; Scheiberg 1990). Family stories offer a form of life writing that draws upon a wide range of sources and a shared set of practices in memory-work, historical and genealogical methods. I came to the story late, as much of the sequence of these historical events had happened before I was born. Dad accidentally discovered the secret of Ada's whereabouts in the early 1960s. He then reconnected with his mother and became aware that some family members knew Ada was hidden away in the mental institution, whilst others did not.

Because of the nature of the discoveries and the various memories included across generations, this is a non-chronological narrative that moves back and forth in time. It begins with my childhood recollections of being in Ada's company with my family. My direct memories of this time are fragmentary and are influenced by our family's stories and shared memories. The narrative interweaves two aspects of Ada's life: the family's version of her absence, and the psychiatric version drawn from Ada's clinical files. It shows Ada's migration from the city to her husband's rural residence, after they wed, and her mobility between home and the mental institutions in the first six years of committal with trial leave and later, her move to other places of care. Akihito Suzuki (2012) argues that migration ought to be considered when examining links between mental health and institutionalisation.

Schmied et al. (2017) state that for mothers, the links between relocation and mental health in the loss of social support and isolation from friends and family are associated with postnatal depression. These points may contribute to Ada's poor mental health due to the risk of isolation when she moved to a small rural town and became a mother, and later her mobility between places of care without social support, particularly acute when her husband kept her whereabouts a secret; this estrangement made an impact both inside and beyond the family. These different migrations are important in the context of a wider discussion about the impact of dislocation on mental health (McCarthy and Coleborne 2012).

Ada's files have provided the family with previously unknown information concerning her diagnosis, the treatments she underwent, and her letters that show her symptoms on full display. On the other hand, the family stories partly fill out the picture, presenting the ways that my family coped with Ada's mental illness, the secrecy around it, and her unexpected discovery. Family stories intertwine with archival sources and letters to understand the ways that Ada, following her institutionalisation, was shunned, and alienated from her children. A major omission in Ada's files is the lack of information about the welfare and care of her children, which this research further seeks to address.

Unpacking the origin of the family secret can give us clues about how my family tried to protect Ada's children and escape the stigma and shame associated with her mental illness. Yet finding Ada's whereabouts has also exposed the long-held secret, damaged family relations and created mistrust. As Barnwell (2019) argues, 'family secrets can be protective or destructive,' and in my family's case, both occurred at the same time. Christine Sleeter (2016) states that critical family history reminds family historians to situate families within larger contexts to interrogate national narratives and identify power relations across socio/cultural groups. This critical family history approach has allowed me to link this family narrative to wider cultural processes and the unjust power relationships between mothers, their families, and mental institutions during the mid-twentieth century in Australia.

## 2. Family Histories

Professional and academic historians, including Babette Smith (1988), Victoria Haskins (2005), and Tanya Evans (2011), have investigated family histories and their potential. In family research, investigations can uncover unexpected and surprising scenarios, such as alcoholism, gambling, cruelty, and adultery. These can shock or upset families and potentially permeate family stories across multiple generations (Green 2013, p. 393).

On the other hand, family members can be grateful when long-held secrets concerning their ancestors' mental illness were unwittingly revealed by researchers (Clark et al. 2022).

Far less work emanates from a researcher's own family, although academic historians have entered the field. For example, historian Graeme Davison, in *Lost Relations: Fortunes of my family in Australia's Golden Age* (Davison 2015), has acknowledged the moral dilemma in deciding whether to include a family secret that breaks the code of family loyalty versus the need for accuracy as a historian.[1] Other historians have reflected on the dilemma of whether to disclose or conceal family secrets when writing on mental illness within their own families. For instance, Lynette Russell chose to reveal her family's secrets and

welcomed the chance to set the record straight. Yet, by 'outing' her family's aboriginal heritage, Russell was subject to lateral violence, unjust criticisms, and the loss of some family members (Russell 2002, p. 8). Possibly a more 'shameful' secret in her family's eyes—one that some family members preferred to keep hidden—was Russell's discovery that her great-grandmother, Emily, had been committed to several Victorian mental hospitals during the 1920 and 1930s (Russell 2002, p. 7).

Micro-histories undertake a deep investigation of one family member. These provide us with what Lindy Wilbraham calls, in her examination of her great-grandfather Harry's mental breakdown and committal, a 'genealogy of the intimate' (Wilbraham 2014). Furthermore, micro-histories concerning researchers uncovering their mentally unwell family members include Marsha Hunt's biography of her paternal grandmother, Ernestine, concerning themes of racism and skin colour (Hunt 1997). After spending fifty-two years in several mental institutions in America's southern states, Hunt discovered the elderly and mute Ernestine living in a poor boarding house. Her medical records lacked information from her husband and omitted that Ernestine was the mother of three sons and that she was also a person of colour. It shows that an archive does not always 'capture the whole truth' (Wilbraham 2014, p. 185).

Commonly, essential facts are left out or taken for granted in mental patient files, especially the existence of children in mothers' files, their welfare, as well as the family's experience of losing a family member to a mental institution. Whether children were dispersed among other family members, boarding schools or orphanages, many families often hid the truth about their biological mother as Ernestine's files attest. This is also the case in Karen Collins' (2021) work on her great-grandmother, Minnie Eleanor Eason. Karen found that Minnie's three children understood that their mother had died when the youngest was a baby. Their descendants were unaware that Minnie was committed as insane in 1917 and had died much later in 1929 at Kew Hospital for the Insane in Melbourne. Both Hunt's and Collins' family histories identified that patient files left out the children and, at the same time, families buried the mother's mental illness and whereabouts beneath secrets (Hunt 1997; Collins 2021). I have also discovered similarities in Ada's files and our family stories (Watts 2015).

Two other in-depth stories by Biff Ward (2014) and Drusilla Modjeska (1990) examine how their families were affected by their mother's mental illness. Modjeska's work *Poppy* mixed fiction with biography and ascribed her mother's silence, depression, and hospitalisation to her isolation as a married woman with children within a middle-class English family in the 1950s. In Biff Ward's book, *In My Mother's Hands*, the author examines her traumatic childhood memories of growing up with her mentally unwell mother. In many ways, Ward's experiences are an alternative story of a husband who tried to keep his wife out of psychiatric institutions, which had dire consequences for their children. These are some examples of 'new family history' where family researchers often shift away from what was once taboo and silenced episodes of family life and instead are willing to disclose mental breakdown, separation, and loss within their own families (Davison 2000). They show the varying degrees in the disclosure of family secrets and different approaches to using real names or pseudonyms by family historians from inside and outside the university system.

I also faced ethical dilemmas when weighing the risks and benefits of revealing family secrets. I used oral history approaches as a method of inquiry and conducted unstructured, conversational, and interactive storytelling borrowed from family historians, family folklorists, and genealogists (Miller 1997, p. 40; Corbin and Morse 2003). Even so, ethical protocols are challenging to apply to humanities and social sciences projects, especially when revealing the lives of intimate others on sensitive topics such as maternal mental illness in the family (Doyle and Buckley 2016; Ellis 2007).

Other researchers have discussed the difficulties in predicting the risk/benefit equation before their study commences, intended to mitigate potential risk in the future (Larossa et al. 1981; McGowan 2020, p. 4). In this research, I have replaced names with pseudonyms

and de-identified place names to protect the privacy of family members, which is especially important when disclosing family secrets. Nevertheless, this strategy may not fully guarantee their anonymity when an occasional long tract of dialogue allows the family to speak for themselves and could reveal the subject's identities (Larossa et al. 1981; Hertz 1997). Changing their names and locations was less than satisfactory for some of my family members who are unfamiliar with this ethical practice in qualitative research. Yet, these ethical protocols are in place to manage risk and protect research participants from harm.

Notwithstanding this, some family members felt that their anonymity made the work less truthful and lacked historical accuracy as a family history. As Wiles et al. (2012, p. 41) point out, their desire to be seen and heard was taken from them. Family names are changed, but as the author, I publish under the family name I share with my ancestors. This shows that the family could be identified despite the ethical protocols in place. This dilemma highlights the tension between sharing authorship with family members under our real names as a compassionate history, which can sit uncomfortably in the scholarly context of 'author-attribution-impact' and feeds research metrics where authors' names are not anonymised.

By comparison, the genealogical approach taken in Wilbraham's family history (Wilbraham 2014), which 'names names', creates an authentic and intimate family narrative. Lindy Wilbraham is less concerned with exposure or risk as she uses her family name 'Wilbraham' throughout the family narrative when tracing her great-grandfather Harry Walter Wilbraham's illness and committal into Grahamstown Asylum, South Africa. The drawback of replacing names and places obscures historical facts and thwarts other family researchers from using them as evidence when tracing ancestral lines in search of living descendants. This has a distancing effect, especially when our side of the family lost contact with Ada's family of origin, her siblings, and their offspring.

Despite such ethical dilemmas, my family were supportive of this project and willingly shared their memories with me as collaborators, not the researcher and the researched, but rather as Portelli (1990) and Kikumura (1986) describe a joint venture between equals. This is not to say that family research is not without risk. And yet my family showed courage in wanting to take that risk and learn what had happened to my grandmother Ada. As I shape Ada's life story into a narrative, drawing upon family memories and Ada's patient files, the research sheds light on both institutional practices and family secrets that alienated our family members from each other, and it is particularly poignant when it was families who played a key role in committing their loved ones into psychiatric institutions. (Finnane 1985; Clark 2018; Coleborne 2010; Watts 2015).

### 3. Childhood Memories

As a young child, I remember waiting with my family for Ada in the foyer of a boarding house in Melbourne. This memory is like a single snapshot of a dimly lit, wood-panelled foyer and marked by the tension of my two brothers being scolded for their noisy behaviour while waiting for Ada to join us for a family outing. Another time, I recall Ada sitting in the lounge room of our family home, smoking cigarettes. I tried to engage her, but she would not smile, and her face and eyes lacked expression. We regularly visited Dad's father, Pop, Ada's husband, on weekends in the Victorian countryside. Pop lived alone in a timber cottage with broad verandas and lead-light windows. Our family visits often involved sending the children outside to play and leaving the adults to talk. During one visit, my parents and Pop settled into the lounge room, where an upright piano stood in the corner. As I headed down the hall to join my brothers outside, I recall one of the adults saying, 'she used to play that piano'. That struck me. I wondered who 'she' was? It was never clear to me as a child that Pop and Ada were my father's parents or why Pop lived alone while Ada lived separately in a boarding house in the city.

I knew there had been something wrong with Ada. I am uncertain when I became aware that she had been removed from her children and locked away. As my father grew to know her more, stories of her absence from his childhood emerged. It was impossible

for me as a child to understand that Dad was getting to know his mother as an adult and parent with his own children. For a long time, I had the sense that Ada had problems related to her menstrual cycle and that once a month, her periods threw her off balance, and she was placed in the sick ward of the mental hospital for one week out of every month. The notion that being female caused her sickness gave me some foreboding as a young girl.

When I was eleven, I accompanied my father shortly after Ada died. It seemed forever that I waited for him in the car, the sun burning through the glass. Dad returned and placed that small brown suitcase on the back seat. That same suitcase later made its way into my childhood bedroom. He said, 'not much for a life.' It was the total of his mother's personal effects.

## 4. Family Recollections

We asked Dad to retell the story of discovering his mother when our family came together in Melbourne in 2006. We were relaxing at my brother's home, and although I had heard the story in various forms before, this time I took notes. Dad had been raised from a baby by his paternal grandmother, Gran. Pop placed Dad's older sister Hannah into the Melbourne boarding school for girls, Lowther Hall, at age four. The children later were reunited when Hannah returned to Gran's care after Lowther Hall closed and was requisitioned for American requirements in 1942 (Darian-Smith 1990). Dad grew up assuming that his mother was no longer alive.

In his early twenties Dad married my mother and together they lived with Pop. Soon after, they built a new home in the outer suburbs of Melbourne and notified their change of address with the local Post Office. Dad received a birthday card redirected from his former address. It was simply signed 'Mother'. It meant nothing to him and thought it must have been one of his mates playing a trick on him. Dad challenged his friend about the card when he visited their new home, but his friend emphatically denied anything to do with it. Unable to reach Pop on the phone, Dad called his favorite Aunt, Pop's sister and asked her about this mystery card. She said: 'I've told your father, time after time, he must talk to you and your sister about your mother' and hung up the phone without any further explanation. Dad then drove to his father's house determined to get to the bottom of the situation. He waited a couple of hours, his mind swirling, before Pop returned home. Finally, he questioned Pop about the card. Pop admitted that it was a genuine birthday card from his mother and that Ada had been living in a mental institution all this time. Pop then produced a bundle of letters and cards that Ada had written to her children over the last twenty years and burned them in the fireplace that day. Pop was instrumental in keeping Ada a secret from her children.

The secret about Dad's mother was out. Ada was alive and living in a Melbourne mental hospital. It is difficult to understand why Pop had kept Ada's letters all this time and then, in such a dramatic way, ruined the chance for Dad to read them. One can only guess that Pop had intercepted and confiscated all Ada's letters to her children to protect them from their mother and the stigma of insanity in our family. He kept Ada a secret from his children, and it is likely Pop intended to continue Ada's long estrangement if that birthday card had not slipped into Dad's hands. Yet her letters and cards do demonstrate that Ada never forgot her children. Cathy Schen's (2005) study of mothers who leave their children shows that from the mother's perspective, ongoing communication with their children can mitigate the effects of prolonged separation and is evidence of their strong-maternal attachment. In this way, Ada had taken the time and care to remember the dates of their birthdays, maintain familial ties, and express affection for her children as their mother.

## 5. Making Contact

Dad soon contacted the mental hospital and met his mother for the first time. He recalled his first meeting with Ada in the early 1960s at Mont Park mental hospital in

Bundoora, Melbourne. He waited to meet Ada in a small walled garden, and was told she would not be much longer:

> I was nervous and unsure what she would look like and what to say to her. I waited for twenty long minutes, and my anxiety increased. When she opened the door, there she was. But six people followed her, to take a look at her son. I didn't know whether to give her a kiss or give her a hug. The first thing she said to me: 'You look just like your father'. It was the last words I wanted to hear at that moment. (Dad, personal communication)

As Pop had hidden his mother from him, Ada's comment that Dad looked just like his father had a painful sting. It was the first of many visits together. Dad recalled taking us to collect Ada from Carmel House, the same boarding house I remembered above, and took her on outings and occasional weekends to our family home:

> When we went to pick her up from Carmel House, she was always late and I remember waiting in that timber paneled foyer, looking searchingly at the stairs, until she made her entrance. When I was working in the city, Mum used to go and get her hair done on her days off, and we would meet afterwards at the Town Hall corner, and go off to Coles Cafeteria, her favourite eating place. But then she would go into the toilets and wash her hair out, so sometimes when I met her, she looked a bit wild and woolly, like she had been diving in the Yarra River. She certainly had her eccentricities. (Dad, personal communication)

As Dad and Ada got to know each other and tentatively forged their new relationship, it was still unclear why Ada was committed to a mental institution in the first place. Dad and his older sister Hannah had little contact or knowledge of Ada's family of origin. They were surrounded by the silence of his mothers' whereabouts throughout their upbringing until Ada's card arrived for him that fateful day in his twenties. We wanted to understand the events that precipitated her committal and found we could access Ada's mental-patient files by applying for a Freedom of Information request. Dad and I discussed it, and he agreed as her next-of-kin. I purchased Ada's birth and wedding certificates to ensure I requested the correct person's files by supplying her full name, maiden name, and birth date. I found that Ada was the younger sister of five brothers. Her marriage certificate stated her occupation as 'comptometer operator' when she married aged twenty-one in 1934. Her employment suggests that she completed a specialised business course to learn the comptometer machine, a key-driven mechanical calculator operated predominantly by women (Wootton and Kemmerer 2007). Ada was among the 94 per cent of women employed in clerical positions in the 1933 census that were young and unmarried (Nolan 1992). These employment opportunities during the depression era, created by advances in office technology, excluded married women, and it is likely Ada left her employment when she married my grandfather Pop.

## 6. Family Access and Reading Ada's Mental Patient Files

Institutional records of adults in Victoria are closed for seventy-five years from the year of the document's creation, meaning the first year the person was committed to a mental institution. This closed-file policy prevents the violation of sensitive material in private records that ensures records are closed for the subject's lifetime (Public Record Office Victoria 2013, Fact Sheet 1). Despite the fast pace of digitisation, not all archives are online or available to the public outside that seventy-five-year period. For example, patient case books for patients admitted to the Beechworth Asylum from 1867 to 1912 are open and accessible online to the public (Clark 2018). It shows that the archival release of patient files, which move from closed to open files, does not happen annually or in systematic manner. Victoria's Minister of Health had discretionary power to grant access to closed files from the twentieth century that formed part of my doctoral data collection (Watts 2015). This gave uncommon access to closed files, likely due to ethical approval and supervisor support through the university system as a doctoral student.

I had applied for and received Ada's patient files before enrolling in the PhD program, and I commenced the study due to the richness of Ada's files. Next-of-kin can request their family member's mental patient records through a Freedom of Information request, regardless of the year of their commitment or the seventy-five-year embargo. Again, with consent from my father, we applied for Ada's patient records. After three months, I received extensive documents on her life as a mental patient from 1936 to 1972, including her final years as an outpatient at Carmel House. Ada's records include committal certificates, diagnoses, notes on trial leave, and experimental treatments, totalling 112 pages that spanned over forty years of Ada's life across several mental institutions in Victoria.

Other historians of psychiatry and family historians have drawn on the large quantities of patient files from specific institutions and several institutions to analyse the historical meanings of 'madness' and changes in treatment over time (Wright 2022, pp. 164–66). In the Australian context, Stephen Garton (1988, p. 189) investigated New South Wales patient records and identified that colonial families had experienced 'high levels of violence, conflict, tension, anxiety and depression'. Garton noted the shift from colonial asylums populated by itinerant male workers to women representing higher patient numbers by 1940, often depressed domestic servants or suburban housewives. Jill Matthews (1984) drew upon twentieth-century female patient files, but rather than analysing the meanings of female madness, Matthews focused on gender history and the ideology of femininity.

Tanya Luckins (2003) focus on women's mental patient files to examine female patients' experiences of loss, grief, mental breakdown, and the gendered nature of wartime loss during World War I. Work on insane mothers committed to Fremantle Asylum, Western Australia, by Alexander Wallis (2020), examined how domestic duties for rehabilitation were a form of moral treatment that emphasised female gender roles within the asylum. Catharine Coleborne's *Madness in the Family* (Coleborne 2010) examined 215 patient case files from four asylums in the colonial period to investigate the different roles families played when dealing with mental institutions concerning their relative's psychiatric confinement, especially in the context of a highly mobile society populated by new immigrants. These large-data studies are made possible due to the unrestricted access to nineteenth-century state archives. But the writing of twentieth-century patients is 'almost prohibited' as Coleborne and MacKinnon described it (Coleborne and MacKinnon 2003, p. 4) due to current embargoes that restrict access to twentieth-century mental patient files. For example, the data collections end in 1910 in both Coleborne's and Wallis' work, and for Luckins, the data ended in 1920.

Garton's and Matthews's work, described above (both published in the 1980s and drawn from twentieth-century patient files) would not be possible in today's climate of restricted access (Coleborne and MacKinnon 2003). Current protection legislation to safeguard privacy has led to a shortfall in scholarship about Australian twentieth-century patients and institutions (Garton 2000). In this way, my access to Ada's patient files, perhaps was a unique and uncommon access to files not often available to many other researchers. Her records offer glimpses into Ada's mental-patient experiences of twentieth-century psychiatric practices and institutions, and are interwoven with family stories during their separation, in which there are so few similar examples.

As I opened the files for the first time, Ada's 'headshot' photograph appeared on the first page of her first admission certificate in 1936. This small photograph is typical of photographic portraiture, cropped as a three-quarter pose with her shoulders angled and her gaze focused to the left of the camera's lens. At age twenty-four, Ada is youthful and smiling, wearing a dress with a large white collar, possibly the same clothes she wore when she left home. Is Ada's smile taken as a sign of her insanity? Or is she realising the situation's absurdity, which is how Lynette Russell interpreted her great-grandmother's smirk in the photograph in Emily's files (Russell 2001)? I prefer translating Ada's smile as an automatic response when facing a camera. Photographs became commonly used in patient records when photography became affordable in the late nineteenth century (Quinn 2002). Patient photographs helped institutional staff to identify patients in escape and

re-admission cases ([Rawling 2021](#)). It is our family resemblance that I had not expected when viewing Ada's headshot for the first time. I could see my face in hers with similar facial characteristics: a high forehead, small eyes, and wavy hair. I identified with our physical similarities and found Ada's photograph intimately relatable, as it confirmed our kinship relationship as her direct descendant. This point ultimately deepened my emotional connection with her. Diana Marre and Joan Bestard ([Marre and Bestard 2009](#), p. 6) argue that 'family resemblances are precisely what define family identity. Our shared physical traits, however subjective in my interpretation, provide a sense of continuity over time.

In the early 1930s, Ada worked as a clerk operating a comptometer calculating machine in the city of Melbourne as a single woman. When she married in 1934, Ada moved away from her social contacts and previous employment for married life in a small regional town on the outskirts of Melbourne. Her migration to the location of her husband's family may have placed Ada in a dependent position and reliant on her husband and in-laws for support and social interaction. A year later, in 1935, Ada gave birth to their daughter Hannah. Being a new mother may have amplified any difficulties she experienced with her in-laws or problems forging new friendships within a rural community. Social isolation and loneliness in new mothers due to the loss of former networks, lack of opportunities to create social connections, and feeling over-burdened by the mothering role are key risk factors associated with maternal mental health disorders ([Lim et al. 2022](#)).

Returning to her patient files, Ada's mental health deteriorated following the birth of her second child and she experienced auditory hallucinations. Ada entered the Royal Park Receiving House, Melbourne, in 1936, two weeks after giving birth. The doctor described her as lucid with 'nervous symptoms that alternate between depression and exaltation along with auditory hallucinations.' The cause of her attack is noted as 'puerperal insanity' and connected to giving birth two weeks before her committal. Puerperal insanity is a nineteenth-century diagnosis that links insanity not only to a recent childbirth but also to lactation, pregnancy, and miscarriage to mental illness ([Hogan 2006](#); [Loudon 1988](#); [Marland 2012](#); [Watts 2015](#)). [Marland](#) ([2004](#)) found that in the nineteenth century puerperal insanity accounted for approximately ten percent of admissions in English asylums. Similar low admissions of birth-related committals occurred in Victoria, Australia in the early twentieth century ([Watts 2015](#)). The terms 'pregnancy, parturition and the puerperal state' as causes of mental illness were removed from Victoria's psychiatric reports by Eric Cunningham Dax, the director of the newly formed Mental Hygiene Authority, in his annual report in 1952 ([Report of the Mental Hygiene Authority for the Year Ended 31 December 1952 1953](#)). In today's terms, it is understood as postnatal depression ([Wallis 2020](#)).

Four days following Ada's first admission, her doctor recorded his observations:

> The patient is cheerful and talkative. She says that lately she has had numerous auditory and visual hallucinations. She is quite disorientated as regards time and place. Her answers to questions are quite irrelevant and she smiles and talks cheerfully for most of the time. (Ada, Mont Park Patient Clinical Notes, AS/1994/508/129)

I found it upsetting that puerperal insanity was associated with dangerous mothers and infanticide ([Marland 2004](#)). It is a dark and taboo subject that conjures the 'madwoman' trope in our cultural imagination. It is rooted in popular discourses such as the murderous mother, the crazy wife in the attic, or the suicidal mother. The horror of maternal insanity plays on our worst fears when, as vulnerable and dependant young babies, we need our mothers to take care of us.

Within the first two months of her committal, the doctor described Ada as 'restless, apathetic, and erratic in behaviour.' The strong links between childbirth and subsequent changes in Ada's behaviour led to the diagnosis of puerperal insanity and her committal in 1936. Yet her files lack information on the length of her labour, delivery method, and the health of the mother or the newborn child. Ada had given birth at a time in Victoria when mothers faced severe health risks during childbirth due to poor standards in obstetrics.

The misuse of interventions and the slow adoption of hygienic practices meant that caesarean section and use of forceps could lead to infection and death, resulting in unacceptably high maternal mortality and morbidity in Victoria (Marshall Allan 1929). It suggests that Ada's labour may have been a negative experience, whether a difficult or protracted delivery, that may have contributed to her deteriorated mental health (Marland 2012, p. 79). Mothers' clinical files rarely included details of birth events, despite adverse maternal outcomes and the strong causal links psychiatrists had identified between puerperal insanity and childbirth (Wallis 2020; Watts 2015).

A lack of improvement in Ada's condition prompted her transfer to Mont Park in February 1937, a large hospital for the insane dedicated to long-term patients located in Bundoora, Melbourne. The doctor's examination on her arrival states:

> Answers irrelevantly. Disorientated as to no place or person. Laughs for no apparent reason. Admits to auditory and visual hallucinations but her statements are unreliable. Nurse says she is impulsive—has to be dressed and led as well.
>
> (Ada, Mont Park Patient Clinical Notes, AS/1994/508/129)

Nine months after her original committal, Ada went home on trial leave and, by March 1938, she was fully discharged as recovered. Ada spent the next two years at home with her family until she became pregnant with their third child. Unfortunately, Ada's behaviour became erratic during pregnancy, and with consent from her husband, she underwent a 'therapeutic abortion' and sterilisation.

> Ada became insane after the birth of a child three and a half years ago. She was then transferred to Royal Park and later to Mont Park. A number of months ago she was pregnant. Therapeutic abortion was performed, and sterilisation was performed by removing a portion of both fallopian tubes. Her mental condition was because insane during her pregnancy hence the abortion, after proper consultation with husband. (Ada, Admission Warrants, AS/1994/114/41)

I found these procedures a disturbing part of Ada's files. The finality of these invasive procedures ended Ada's reproductive life at the age of twenty-eight (Watts 2021). Her history of puerperal insanity likely provided the psychiatric grounds to proceed, but it did not bring her peace of mind. In May 1940, soon after the therapeutic abortion and sterilisation, Ada was re-committed back into the mental institution:

> Well-oriented as regard time and space. Talks quite rationally. Says that for the last 12 months she has been hearing voices when there is no-one about; it seems as if her thoughts are being spoken aloud. Does not get on well with husband. She is the youngest of a family of six. (5 brothers)
>
> Two months later:
>
> Little change. Is apathetic and indifferent about her fate. Reluctant to go home again. Says does not get on well with home and appears to have no love in regard to him.

Ada expressed to the psychiatrist how she felt about her husband, and we start to gain insight into their marriage difficulties. Guilt and remorse from the termination of her pregnancy may also play a role in her apathy, as Jeffcoate states that psychiatric trauma can occur in some women who underwent therapeutic abortions (Jeffcoate 1960). Despite her indifference and problems with her husband, Ada left for a ten-month trial leave, showing her mobility between institutional care and the family home. On her return to the institution in May 1941:

> Patient is a young lady in good physical condition. She talks freely with some affectation of home. She has various delusions about her husband, thinks he is jealous of her listening to wireless. In addition she says he keeps taunting her about her mental illness. They are on the verge of divorce but husband wants to claim the children. She has complained to police on many occasions mainly

about her husband but she has been unable to get any satisfaction—feels that
police force is corrupt. Admits auditory hallucinations, still hears voices over
wireless, flight of ideas, delusions and hallucinations. (Ada, Mont Park Patient
Clinical Notes, AS/1994/508/129)

It is difficult to assess if Ada was delusional regarding her husband or whether she
experienced domestic abuse and was subject to her husband's coercive control and sought
police help. Chances are that her husband would deny these accusations if questioned. Yet
there is a history of policing that turned a blind eye to domestic and 'private' complaints
by failing to assist when requested (Piper and Stevenson 2019). There may be more
family secrets here that are impossible to tease out. Still, Ada's auditory hallucinations are
taken as a sign of her 'insanity,' her fears dismissed as delusions, which undermine her
ability to remain outside the mental institution. At Mont Park, Ada's diagnosis changed
from puerperal insanity to primary dementia in July 1941. Dementia, and dementia
praecox, at that time, were diagnostic terms created by Emil Kraepelin that described the
degeneration of cognitive faculties (Berrios 2005). There are no medical notes that depict
Ada's reduced cognition.

On the strength of this change in diagnosis, Ada underwent a series of insulin-coma
treatments (ICT) for the next three months. ICT involves giving patients daily injections
of insulin to induce hypoglycemia and coma. Manfred Sakel developed ICT in Europe
and it was implemented by Dr Farran-Ridge and Dr Reynolds at Mont Park (Kirkby 1999;
Kaplan 2013). Under the supervision of Dr Reynolds, Ada showed 'marked improvement'
following ICT every morning for three months:

Mentally much improved—states she is no longer hallucinatory although remem-
bers that she was so and seemed to have gained full insight into her condition. A
satisfactory remission. (Ada, Mont Park Patient Clinical Notes, AS/1994/508/129)

ICT appeared to aid Ada's recovery and ended her auditory hallucinations for the
first time, but these effects did not last. It proved a dangerous and experimental treatment
when some patients died due to complications (Watts 2021). Ada experienced a short-lived
remission until she returned to the institution suffering grandiose delusions following six
months of trial leave at home. It is hard not to believe that going home on trial leave, a
form of mobility considered helpful in some instances, contributed to Ada's worsening
mental health. Her doctor described her delusions of grandeur in January 1942:

She also stated—you are an English Doctor, I am Oxford graduate myself. Dr
F.E.M ie. Female doctor. Also an Insane specialist not practicing and an RPM
practising. For a long time she would not disclose significance of RPM but
eventually stated meant Royal Pedigree M [ . . . ] and referred MO to enquire
from Churchill and British Secret Service.

She claims she arranged the introductions between Duke & Duchess of Kent. She
is willing to oblige for a fee of 1000 pounds to do same for the royal personages.
She stated this country is now known as L'Aurolia Republic—it was Australia
prior to the revolution in 1900. She says she is a member of Russian Royal
family and her real name is Lily Vertel Rose Alvaradora Icebel and says herself
'Alvara Russia.'

She has numerous other fantastic and bizarre delusions. Naturally she is some-
what exalted. She admits hearing voices talking about British Secret Service work
so cannot divulge subjects discussed.

She is cheerful and cooperative, her mood is cheerful, but she is living in a world
divorced from reality. (Ada, Mont Park Patient Clinical Notes, AS/1994/508/129)

Ada is in an exalted state with claims of prestigious professional status, medical
accomplishments, and of Russian royal birth. Victoria Shepherd defines a delusion 'as
a fixed, false idea, not shared by others, unshakable in the face of decisive evidence
contradicting it (Shepherd 2022, p. 10) Richard Noll (2007, p. 121) adds the grandeur aspect

to delusions 'in which a person is convinced that he or she is 'special' due to an inflated sense of power, identity, wealth, or relationship to a deity or special person.'

Ada's grandiose delusions are aspirations that bolster her self-worth and achievements: she can arrange royal introductions, is born into the Russian Royal family, and works for the British Secret Service. Knowles et al. (2011, p. 688) argue that such grand delusions may operate as a defence mechanism and, in Ada's case, may ward against feeling like a failure in losing her identity and position in the family as wife and mother. It is made worse by the humiliation of being reduced to the small and narrow confines of a mental patient, possibly threatening her self-esteem. Yet Ada's grandiose delusions did not operate outside of culture. She was committed again to the mental institution in early 1942 when Australian troops fought in campaigns across Europe, the Mediterranean, and North Africa as part of the British defence in WWII. Her reference to Churchill and her work with the British Secret Services illustrates how Ada's grand delusions were shaped by Australia at war and provides glimpses of the possible political anxieties and the turmoil of war fuelled by wireless news bulletins and newspapers of the day.

Regarding Ada's reference to the Duke and Duchess of Kent, Princess Marina is a descendant of the Russian royal family, and she married Prince George of the British monarchy the same year of Ada's wedding in 1934[2]. This royal marriage was highly publicised, and the first royal wedding was broadcast live on the radio to the public (Owens 2019). The media spectacle emphasised the high fashion, romance, and glamour that appealed to young female audiences, such as Ada, who was twenty-one at the time. Marina's heritage may have inspired Ada's grandiose delusions in ascribing to the same background as a member of the Russian royal family. In addition, Russia became Britain's ally following the German invasion of the USSR in 1941. These are some possible ways which the royal events in the British monarchy and the Australian campaigns during WWII influenced Ada's delusions of grandeur.

Further examples of her grandiose delusions are evident in two letters Ada wrote and held within her files. One letter is an application Ada penned to join the Women's Royal Australian Naval Services (WRANS), dated 3rd January 1943. In 1942, WRANS commenced a large-scale recruitment drive seeking to employ women to help with increased naval demands in the Pacific (Christopherson 2010). Ada responded to the WRANS advertisement in *The Argus* newspaper on 18th December 1942, which sought to enlist 300 women into the newly formed naval service (*The Argus* 1942). For example, they required female applicants for telegraphists, signallers, car drivers, supply attendants, writers, and cooks. In Ada's WRANS application, she offered her medical services as a trained doctor and surgeon and noted she would be available for her naval commission in three to six months. Again, she fabricated her educational credentials, and her Russian royal birth claim persisted. Even so, Ada wanted to be helpful in war service and gain employment, as she had done so in the past, just like the other 1000 women who enlisted with the WRANS by the end of 1942 (Dennis et al. 2008, p. 607). The address Ada provided for return correspondence is the Female Ward, Mont Park Mental Hospital, which indicates she had some grasp on her present reality in giving the correct location. The second letter, held in her files, is addressed to her husband, my grandfather, written in 1943:

> One of the patients here has been friendly and helps to pass the time. With your consent, we are planning to go to England together, as I must see to my English affairs, being of Russian titled birth. I inherited from my father about twenty-six years ago, various properties in England, America and Australia, also Europe, these properties, and affairs await my attention in London.

> Also my Army Commission, with the British Secret Service, which commission continues from my last visit to London (not having been notified to the contrary), including secret service to England and America during this war, also been specially called into commission in about 1938–39. That army report I must make in London at a convenient time, and visit my regiment.

> And lastly, my enormous will, I must change it to include yourself, my husband, and only two children, and those several titles to descend upon both, also a ducal title upon yourself from marriage (to me), now Alva.

> I should be pleased to have your cooperation about my proposed voyage, and sympathy. That commission mentioned, being partly executed under your roof, I should be honoured to share that salary, and tender my apologies for the inconvenience.

> In London, I have my own staffed residence waiting at Westminster, which matter I shall enquire before leaving Melbourne. (Ada, Mont Park Patient Clinical Notes, AS/1994/508/129)

Here, we find greater detail of her grandiose delusions comprised of extensive overseas properties, travel to London necessary to complete her British secret service mission and updating her will to confer her wealth and titles to her husband and children. It is an interesting convenience that Ada could not disclose the nature of her secret service work. After all, possessing a hidden talent or undertaking clandestine activities would make it difficult to disprove or contradict. Spies are sworn to secrecy, and is it that the mental hospital is an ideal hiding place for a spy to remain incognito? Ada certainly didn't see herself as any ordinary mental patient. Instead, her 'delusions of exceptionality,' as Isham et al. (2021) termed grandiose delusions, illustrate her desire for social mobility and increased status by casting herself as a wealthy Russian royal working as a British spy.

Letter writing is usually a social practice when there is an exchange between people (Barton and Hall 2000). However, in this instance, the institution withheld Ada's letters, which allows us access to her state of mind. Cawte (1998, p. 42) remarks that at Parkside mental hospital, the Superintendent read all patient letters and decided to either withhold or send patient letters. We are not the intended reader of Ada's letters, and they illustrate her sense of self, presence, and distinctive voice. Her tone is not friendly, casual, or intimate towards her husband. Instead, Ada's grand delusions are on display and presented in a formal and authoritative voice. In a study of individuals with schizophrenia, those experiencing grandiose delusions were more optimistic regarding their future and this provided incentive to maintain and strengthen their grandiose ideas (Bortolon et al. 2019). Shepherd argues that delusions help to assimilate a dramatic loss in status in a life that went catastrophically wrong and to reconcile a fallen and lowly existence (Shepherd 2022, p. 10). Ada's delusions of grandeur are a preferable alternate reality that reconciles her fall from grace, the loss of family, the social inequality and the powerlessness of her life as a mental patient. We can see this in Ada's willingness to share and bestow her enormous will, land, and titles upon her husband, which subverts her husband's power with her own. After all, it was his position and resources that sought her committal. Instead, as a wealthy royal, Ada is in a position of power and privilege, not her husband. Isham et al. (2021) interviewed patients about their experiences of grand delusions. They described their grand beliefs as highly meaningful, providing a sense of purpose, and belonging, or making sense of challenging events. Acting as a highly educated spy, Ada had given herself a wartime job. This gave her ambitions direction and a sense of purpose as a patriotic Australian and loyal subject to the British Crown and the Commonwealth during wartime, as the WRANS application attests. Shepherd (2022, p. 50) reminds us that by listening very closely, we can come close to understanding the people experiencing delusions. By actively listening to Ada's grandiose delusions, I hear her demand for dignity and respect as her response to the injustice she suffered, the loss of her autonomy, power, freedom, and the ability to mother her two children.

Unfortunately for Ada, displaying her grand delusions meant no further trial leave following her return to Mont Park mental hospital in 1943. Ada spent the next twenty years in several mental institutions in Victoria without trial leave, often a pathway to discharge for patients with supportive families. Her files are brief throughout the rest of the 1940s and mid-1950s following Ada's grandiose delusion letters. In 1948, her doctors note that Ada was 'considerably improved, is much quieter, works in the wards and is no trouble in

any way.' A further entry states that she is a 'delusional schizophrenic whose moods are variable.' This is the first time 'schizophrenic' has been used in Ada's files. By 1954, Ada is transferred from B ward to A ward and 'works daily in B ward, which she calls going to the office. Some delusions. Wishes to go home and finish her studies at the university.' The mention of her ward work as 'going to the office' shows her sense of humour and a reminder of the days she worked in an office as a comptometer operator before her marriage. Furthermore, her desire to complete university point to her desire for social mobility through educational aspirations.

Ada worked as a domestic servant for the Donnan family at Mont Park. After migrating with his family from England, Dr Donnan took up the medical officer position at Mont Park in 1955. Interviews with Dr Donnan's youngest daughter Sally provides a valuable personal dimension to understanding Ada's life as a mental patient. Sally recalls conversations with her older sister:

> I don't recall Ada at Mont Park, my sister remembers that she and I got sick while we were living at Mont Park, it was the measles she believes, and Ada came in to help mum look after us. My sister remembers she was very sweet, and in a bit of a fluster. So I am sure that was the introduction of Ada into our household. It would have just gone on from there, with Ada being a help to mum and helping with the household routines. (Sally 2013a, personal communication)

Dr Donnan was promoted to Chief Superintendent Psychiatrist at Beechworth Mental Hospital in regional Victoria after one year at Mont Park (Victorian Government Gazette 1956). Sally recalls traveling with her family and Ada from Melbourne to Beechworth for her father's new position:

> There were five of us, including my elder sister, driving up to Beechworth in Dad's Alvis vintage car with lots of luggage. It got dark and none of us were that comfortable being in the middle of the bush at night. The car conked out at a river crossing, water was running over the road, in the dark. Ada was sitting between us two kids in the back, gripping our hands. Anyway we got going again, though I don't remember the rest of the trip. (Sally 2013b, interview)

The Beechworth mental asylum was built in 1867 and based on the Colney Hatch asylum, England, of Italianate design with a three-story administrative building on 200 acres (Craig n.d.). The Donnan family settled into their new home located within the grounds of the mental hospital, where Ada had a bedroom in their home and worked as their domestic servant. Unpaid work by patients was widespread and drawn from Tuke's moral therapy and considered therapeutic for patients. Patients' work was crucial to the mental hospital's economic management and a form of therapy for patients intended to promote recovery (Coleborne 2003). Patient activities are divided across gendered lines: male patients did farm work while female patients undertook work in the laundry, sewing, and cleaned wards (ibid.). Sally recalls Ada and her mother working together in their home:

> Mum and Ada were perfectly aligned with routine. A match made in heaven. Everything was on time, the routine, brushing hair, cleaning shoes, preparing school uniforms and meals. I seem to remember Ada being dressed very neatly every day, usually in a black skirt and white shirt. She wore her hair in a well cut bob. I always remember her wearing thick beige stockings and stout black shoes. Pink complexion, blue eyes. She never wore makeup. One day mum told me she had a son, which made me wonder a lot. (Sally 2013a, personal communication)

Sally provided several pieces of information I found fascinating. Her description of Ada's physical appearance with 'pink complexion and blue eyes' are familiar traits several of us share in our family. Her neat attire, black skirt, and white shirt mark her class and lower status as a domestic servant and are used to create the distinction between the head of household and the maid of the house (Louvier 2019). The mention that Ada had a son,

my father, is surprising. Dad met Ada after she had left Beechworth, which shows that others knew more about Ada's family while others remained unaware.

By 1963, Dr Donnan transferred from Beechworth to Brierly mental hospital in Warrnambool, Victoria. This time the Donnan family could not take Ada with them to their new posting as they had done in the past. There are several letters in Ada's files concerning what was to become of Ada once the Donnans left Beechworth. Pop, my grandfather and Ada's husband, wrote to Dr Donnan to ask if Ada could manage at home. Pop was worried that if she came home and 'slipped back,' she could cause trouble with our children, who were both married with young children of their own (27 November 1962). At this time, Dad did not know about his mother. If Ada came home, there was more at stake than Ada 'slipping back.' Pop would have to explain her return and where she had been all this time, exposing his role in Ada's concealment. As it turned out, Dr Donnan replied that Ada was incapable of coming home, as she could not manage the house, purchase food, or cook it properly and continued to insist she was of royal birth. He recommended that it was in Ada's best interests to transfer to Mont Park mental hospital, where she could live a simple routine (30 November 1962). Ada moved back to Mont Park in January 1963 when the Donnan family moved away. Dr Bozan, Deputy Superintendent at Beechworth mental hospital, wrote in Ada's transfer letter:

> Mrs Ada [ . . . ] was transferred from Mont Park Mental Hospital to Beechworth on the 26-7-56. She spent all her time working for Dr Donnan. She worked quite satisfactorily and seemed very happy with Dr Donnan and his family. After Dr Donnan left the hospital she is missing the family very much and would like to return to Mont Park. Her family lives near Mont Park and could visit her quite often. (9 January 1963. Ada, Mont Park Patient Clinical Notes, AS/1994/508/129)

I imagine the change from living and working with the Donnan family at Mayday Hills, Beechworth, to being placed in the female ward at Mont Park was a challenging transition for Ada, another 'migration' in her experience of mental health treatment. The Mont Park doctors' assessment of Ada is brief: 'no medication, physical abdomen scar, cooperative, talkative. Delusions of grandeur which she voices quite casually in a flat, emotionless voice, good worker'. It is interesting to note that Ada's abdomen scar is likely to be either the result of a caesarian birth or the tubal ligation performed over twenty years ago. She was not receiving medication as a schizophrenic patient, and her grandiose delusions did not appear to interfere with her work ethic. It was around this time that Dad first met Ada, anxiously waiting for her in that walled garden at Mont Park.

## 7. Carmel House

A year and a half later, in 1964, Ada moved again. She transferred from Mont Park to Carmel House, a residential boarding house for female patients in Preston, Melbourne. Dr Eric Cunningham Dax, then the head of the Victorian Mental Hygiene Authority, reported on the acquisition and purpose of Carmel House:

> This is an old mansion surrounded by beautiful lawns and gardens, and with a residence for the psychiatrist in charge. The building was being used as a private hospital known as Carmel. It is anticipated that this will be used as a hostel for some twenty female patients. (Report of the Mental Health Authority for the Year Ended 31st December 1963 1965)

The Mont Park doctor assessed Ada in preparation for her transfer to Carmel House:

> Patient is schizophrenic and is quiet most of the time. Occasionally deluded. Excellent worker. No medication at present. Arrangements made to board her out to Carmel House. She is anxious to go. (20 August 1963, Ada, Mont Park Patient Clinical Notes, AS/1994/508/129)

Ada lived at Carmel House, worked as a domestic servant, and regularly met my father for lunch at Coles cafeteria in the city. It was here at Carmel House, too, that my earliest childhood memories of Ada took place when waiting with my family in the foyer to take Ada on regular visits to our home. Ada attended the Ernst Jones Clinic, Preston, as an outpatient and is described as a 'chronic schizophrenic, with auditory hallucinations and has delusions that she is related to royalty.' These symptoms had been constant and long-lasting for the past thirty years, and it is in this examination that the doctor notes the recent prescription of Stelazine and Mellaril for the first time. These were among the first-generation antipsychotic drugs to treat schizophrenia (Braslow and Marder 2019). Ada's grandiose beliefs persisted well into the 1960s, as my father recalls:

> The first time Mum stayed at our place for the weekend, Mum turned to me and said 'Tell me, do you still use your title?' 'Er, what title's that, Mum?' I asked surprised. 'Lord Barry,' she replied, as if everyone knew. 'No Mum. People don't use their titles much these days.' 'What a pity,' she said and dropped the matter. (Dad, personal communication)

Ada lived at Carmel House until she died in 1972, aged 60. One of Ada's brothers, Alan, who lived in Sydney and had visited with Ada several times when on business in Melbourne, wrote to Dad after learning of her death:

> We saw her regularly when we visited Melbourne some few years ago—took her driving and saw her quite often, but we both felt we could not do much to make life better for her.

> We think she had some happy years at Beechworth where she lived in the home of Dr and Mrs Donnan who had two young daughters. The Donnans are now at Warnambool and I wrote and told them of your mother's death. All in all, Barry, we both think her passing was probably a good thing in all the circumstances.

Alan was the only member of Ada's family who provided her with support throughout her time in various mental institutions. Later he was in contact with Dad, but only after Ada was discovered. This suggests that Alan too kept the family secret of Ada's whereabouts.

By unpacking the silences that shrouded Ada's life, she has become visible and, importantly, is brought back into the family fold. The effect these secrets had on Ada's children and descendants has been presented through stories. Tanya Evans (2020) reminds us that focusing on female ancestors is a feminist approach that challenges the silences around women's lives. This work has followed Ada's journey as a mental patient. I would like to think that the depictions here may be helpful for other family historians. What I have presented are illustrations of the methods and steps required to access the files of family members, particularly those who are next of kin and therefore accessible to family researchers. Nevertheless, one cannot fully prepare for previously unknown information and the shocks and surprises these types of records contain.

It was an arduous and emotional journey. Like Wilbraham (2014), I also found that engaging with the content in Ada's mental patient files and these family stories was somewhat harrowing. It has been tough emotionally coming to terms with the hiding of Ada's whereabouts from her two children. The long-held secret of her mental illness is difficult, and the accidental revelation that found her living in a mental institution is painful. The family lies told to cover it up remain a challenge for our family.

Ada's files were full of difficult revelations for our family and required much emotional labour as we came to terms with the contents. While the various medical treatments Ada underwent also became a point of hope for the family. The discovery of the drastic therapeutic abortion, sterilisation, and insulin-coma therapy was the source of alarm and horror. We marveled at her audacity, intellect, and creativity of her grandiose delusions as a Russian princess and British spy and wondered if our royal status had gone awry. Ada's letters provide the closest insights into her own experiences of mental illness, and by listening closely to her grandiose delusions of fame and fortune, Ada tells us that she wants to be treated with respect, dignity and be triumphant.

Yet neither the archive nor the memories of relatives can answer all the questions the family asked about Ada's first committal in 1936. Were my grandparents wrongly suited partners and caught in a difficult marriage? Was Ada in an abusive relationship? After all, the number of 'housewives' in New South Wales mental hospital populations increased in the 1940s (Garton 1988, p. 147). Did husbands exploit the committal process to 'dump' their wives so they could freely move on with their lives and enter relationships with other women (Scull 1979; Gittins 1998, p. 49)? Was there an abuse of patriarchal power in the collusion between husbands, doctors, and mental institutions to hide inconvenient or troublesome wives? These are examples of the gaps and questions historians must ask.

To examine these questions further, I asked Dad about what type of man his father was. Dad recalled his earliest memories of his father Pop, when Dad lived with his grandmother, and his sister Hannah was boarded at Lowther Hall:

> Pop would come on Sunday's afternoons and evenings. That's when I saw him and we used to go for walks together through the paddocks and he used to lift me up over the fences. Time spent driving with my father are the warmest memories I have of him. Sitting between his knees, my hands on the steering wheel, 'driving' the truck and peering through the wheel for glimpses of the road ahead, shifting my hand on the gear stick with his hand firmly over mine. He taught me to sing 'Little Sir Echo' with him as we drove. It was great fun. But it wouldn't be true to say I was close to him. He was my Sunday father. My bonds were really built with my grandmother and her youngest son as we lived together at Grans' place. Later, when Hannah returned from boarding school, Dad left Gran's house and moved in with Pop, Hannah and their cousin's family. Pop continued to work long hours in his carrier business, returning late after we were in bed. Pop was an even-tempered man, he was never violent, and never disciplined us. But he would withdraw himself from difficult situations. He knew how to detach himself. He never did provide any explanation about why he withheld my mother's letters. I was furious about it. He knew he had done the wrong thing. The longer he left it the more difficult it was to put things right. He had built himself into a situation. I think I recognized he had found himself in a situation he couldn't handle properly as a father. And I also know he was enjoying himself sexually, he had the rights of a free man in his head, despite the fact he was married. I still can't overlook the level of detachment he had about the whole thing.

Pop enjoyed some advantages of being a single father after his wife Ada went to the mental institution. He worked long hours building his truck business and being relieved from the daily demands of child-rearing; he had time to explore relationships with other women. He devoted Sundays to spending time with his children and yet at the same time he kept the secret of their mothers' whereabouts from them. This story of the absence of Ada from the lives of her two children due to her mental illness, is a microhistory that intersects with the larger narrative of standard past practices that alienated families from each other. Her story is one of some mobility: a migration to the regional town where she fell ill, and her movement between institutions, as well as her dreams for greater mobility, evinced in her delusions. Moving, alienation, and removal are all striking themes in histories of mental health and institutionalisation. The ramifications for families from child-removal policies, including Care Leavers and Stolen Generations, share similarities with the mother's removal from her children in asylum committals that fractured families.

We can see that concerns for the children's welfare were not part of the institutions' purview. There is no information in Ada's files stating that she had two children nor how or who would care for them. Marland reminds us that 'it was simply assumed that someone will take care of the newborn in the mother's absence.' (Marland 2004, p. 60). Therefore, the value of family stories illustrates that Dad and Hannah grew up, notwithstanding hardships, to have productive lives and families of their own. The welfare of Dad and Hannah is a crucial aspect left out of Ada's files and is a point of dissatisfaction for our

family. In this way, critical family history places in context the effects of policies, norms of the time, gendered power, psychiatric knowledge (or the lack of it), institutional regimes that are combined with painstaking research by family historians and genealogists to track the ways children of unwell mothers were cared for or dealt with.

The more positive aspect of this troubling narrative recognises that Ada had experienced a sense of family life with the Donnan family and their two daughters at Mayday Hills in Beechworth during the 1950s and 1960s. Dad heard many anecdotes from Ada about the Donnan family events, including a wedding, during their lunches in the city when Ada lived at Carmel house. Dad had urged me to find out more about the Donnan family and I managed to find and interview Sally, who generously shared her memories of Ada living with her family throughout her childhood. I learned to appreciate Ada's diligence and hardworking nature and came to understand that her dreams of fame and fortune may helped Ada to come to terms with life's disappointments, thwarted ambition and failure to achieve upward mobility as a young woman.

Immersing myself in this work has led me to digest and process the past and lessen the grip this family secret has had on my family. Family stories matter because they show what patient files do not, the legacy of family separation and the secrets told. This research is vital in how families and institutions caring for children in the absence of their biological parents manage the truth. In reading Ada's files, we glimpse how mental institutions treated unwell mothers and her movement between institutions. Thus, the secrecy and shame associated with mentally ill mothers meant that Ada was hidden from her children and her son's later discovery more profound. This family loss has impacted the following generation's sense of family identity and teaches us about the ramifications of family secrets and the possibilities of healing when sharing troubling family stories.

**Funding:** This research received no external funding.

**Institutional Review Board Statement:** The study was approved by the Human Research Ethics Committee of SOUTH-ERN CROSS UNIVERSITY (ECN-13-018, 2013).

**Informed Consent Statement:** Informed consent was obtained from all subjects involved in the study.

**Data Availability Statement:** Not applicable.

**Conflicts of Interest:** The author declares no conflict of interest.

## Notes

[1] Other historians currently writing their family histories include Penny Russell, see: Related Histories: Studying the Family, Conference Program (2017) p. 31 and Marian Quartly, see: https://research.monash.edu/en/persons/marian-quartly (accessed on 7 August 2022).

[2] Princes Marina of Greece and Denmark was Greek by birth, whose mother was Grand Duchess Elena Vladimirovna of Russia, a descendant of Emperor Alexander II of Russia.

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
