# Peer review of "Maternal Insanity in the Family: Memories, Family Secrets, and the Mental Health Archive"

_genealogy, doi:10.3390/genealogy7010005_

Round 1

Reviewer 1 Report

See attached file.

Author Response

Thankyou for you helpful feedback and comments.
Please see the attachment

Reviewer 2 Report

The paper would be improved by some information about Ada's parents and siblings. The author has the birth certificate so this information would be there. It would seem that Ada was estranged from them by the time of marriage (or they didn't live in Australia or Melbourne or had all died) and there is no mention of them providing support during her admission. 

The paper is very much from the view of the grandchild.   More about her family - even a statement that there was no contact with her parents or siblings would provide some balance to the story.

Author Response

Thankyou for your helpful comments and feedback. Please see the attachment
